# Mediterranean Extensive Green Roof Self-Sustainability Mediated by Substrate Composition and Plant Strategy

Francesca Vannucchi [1,2], Carlo Bibbiani [3,*], Claudia Caudai [4] and Francesca Bretzel [1,2]

1    Research Institute on Terrestrial Ecosystems, National Research Council, 56124 Pisa, Italy; francesca.vannucchi@cnr.it (F.V.); francesca.bretzel@cnr.it (F.B.)
2    NBFC: National Biodiversity Future Center, 90133 Palermo, Italy
3    Department of Veterinary Science, University of Pisa, 56124 Pisa, Italy
4    Institute for Information Science and Technologies "Alessandro Faedo", National Research Council, 56124 Pisa, Italy; claudia.caudai@isti.cnr.it
*    Correspondence: carlo.bibbiani@unipi.it

**Abstract:** In the cultivation of extensive green roofs (EGRs), substrate composition is a key aspect together with the evaluation of suitable recycled materials. Recycling materials as amendments can improve the establishment of a self-sustainable EGR, thus providing ecosystem services and benefits from a circular economy and climate change perspective. This study investigates the effects of compost and paper sludge on water retention, substrate temperature attenuation and plant diversity in an EGR experiment. The substrates were composed of tephra (V), compost (C) and paper sludge (P) as follows: VC, as control, VPC and VP. Herbaceous species with different ecological functionality (succulents, annuals, perennials, legumes, geophytes) were sown and/or transplanted with no cultivation inputs. Plant community composition -abundance- and diversity-richness-, substrate water retention and temperature were analyzed. The VPC and VC had the same average substrate temperature, with values lower than VP. The water retention capacity was higher in VC, thanks to the presence of compost. The substrate with paper sludge (VPC and VP) showed the highest species diversity. The VPC substrate was the best compromise for EGR temperature mitigation and plant diversity improvement. Plant functional types in EGRs can be increased, and thus the biodiversity, by modulating the quality and percentage of amendments. The substrate composition can also affect water retention and substrate temperature. In addition, the use of recycling paper sludge in growing media is a winning strategy to reduce waste.

**Keywords:** annuals; biodiversity; herbaceous species; nitrogen; water retention curves; recycling

## 1. Introduction

Nature-based solutions are focused on tackling the challenges posed by climate change and the loss of biodiversity: climate resilience, water management, natural and climate hazards, green space management, biodiversity, air quality, place regeneration, knowledge and social capacity building for sustainable urban transformation, health and well-being, new economic opportunities, and green jobs [1]. Nature-based solutions can also improve the urban water cycle, enabling the elements in water to be reused [2]. Extensive green roofs (EGRs) exploit a shallow layer of growing media where plants thrive, often with limited fertilization or irrigation inputs. EGRs help conserve biodiversity and provide habitats for species [3], mitigating heat, and the impact of rainwaters and pollutants [4]. The characteristics of the substrates that are suitable for EGRs have been widely studied in order to find the correct trade off to fulfill the multiple requirements: weight, resistance, allowing plant growth, and an equilibrium between mineral and organic components.

The chemical and physical properties of green roof substrates are important for supporting a self-sustainable plant community, with a different functionality and life forms, inspired by the habitat analogue perspective [5]. Green roof substrates are composed of

a mineral and organic part, with a rate of 70–95/5-30%. The mineral part is generally of volcanic origin material (e.g., pumice and pozzolan) which confers a high porosity and low density, thus the passage of air, which is a better insulator than water. The organic part is more critical because it is the most active: it releases nutrients which affect the presence of plants and biota and it is subjected to shrinkage and degradation. Although the organic component is necessary for providing the plants with nutrients, its high-water retention increases the load of the substrate. For this reason, normally organic part is provided as 5–30% of the growing media [6]. The organic component is also subject to consumption and nutrient leaching, and it is denser than the mineral component. Another issue is that undesired plant species (weeds) may develop, either from the seed bank (compost) or due to natural dispersion (wind, birds, water, insects), and if the availability of nitrogen in the growing media is sufficient, the weeds can develop an unsightly biomass. The organic parts commonly used in growing media are peat and compost from organic wastes [6]. Peat is now limited in its use as organic component, due to reasons related to environmental protection, and compost is considered controversial [7]. Compost from green or municipal waste may not be consistent in its properties, especially if it is derived from urban mixed waste, which is why it is generally not employed in nurseries and agriculture, where coconut coir is preferred [8]. Alternative materials have been studied to improve the EGR substrate properties, limiting the use of peat and compost. In addition, the use of alternative materials (crushed bricks, biochar, glass, cement) is in line with the circular economy concept [6]. However, as yet, there is no definitive solution. Paper sludge (PS) could be a good organic component in EGR substrates. PS has a high pH and carbonate and organic content and is used as an amendment in agriculture, as it increases organic carbon and the cation exchange capacity [9]. Mixing PS with green waste compost improves the growth and transplant of ornamental trees [3]. In addition, PS added as pellets to EGR growing media favors the development of more plant species and functional groups compared to media without PS, thanks to the low availability of nitrogen [10].

The temperature of the EGR substrates is related to the solar radiation, and the insulating contribution of the growing media is determinant, especially in buildings without insulation [11]. The properties of the media are thus key to this process. Regarding thermal performance, self-sustainable EGRs behave differently under extreme drought conditions depending on the plant species and the substrate properties [12]. Sedums have been shown to perform well in terms of thermoregulation all year around in EGRs [13], while other studies highlight better results with non-succulents with high stomatal conductance and leaf area index [14] or gray-leaf plants with high albedo [15].

Most EGR studies in the Mediterranean or in other arid conditions involve experiments that use irrigation, or at least emergency irrigation [16]. Our experimental study focuses to define EGRs' sustainable practices excluding irrigation as no longer a solution due to the climate change drought emergency in geographical areas such as the Mediterranean. The aims of this study were (1) to evaluate the role of EGR substrates amended with recycled materials (compost and paper sludge) in supporting a self-sustainable vegetation; (2) to select the EGR substrate composition that represents the best compromise between substrate temperature mitigation and plant diversity in a Mediterranean climate. Specifically, the water retention and the temperature of the tested substrates were evaluated, as well as their effects on the dynamics of vegetation, in terms of plant richness and abundance and plant functional types.

## 2. Materials and Methods

### 2.1. Experimental Design

The experiment, started in 2014, was carried out on a flat ten-meter-high roof at the Italian National Research Council in Pisa (Italy), (43°43′9.707″ N, 10°25′15.463″ E) as reported in Vannucchi et al. [3,10]. Briefly, twelve ITM (interlock turf module) 1 × 1 m black plastic boxes were set up, with 10 cm depth, and 8 cm of gravel for drainage. The boxes were arranged in two columns and six rows and filled randomly with three different

substrates to obtain four replicates each. The substrates were made up of compost from municipal green waste, commercial tephra product (Vulcaflor) and pelletized paper sludge as follows: VC 80% Vulcaflor + 20% compost; VPC 65% Vulcaflor + 27% pellet + 8% compost and VP 40% Vulcaflor + 60% pellet. The three selected materials were mixed to obtain growing media with about 10% organic carbon. The use of paper sludge aimed to reduce the nitrogen content in the growing media. In addition, the sludge was pelletized to increase the durability, percentage of air in the growing media, water retention, and drainage, and to make it more manageable.

Besides recycling a waste material that was free of pollutants, the aim of the paper sludge amendment was to reduce the substrate nitrogen content, thus reducing the plant biomass and creating gaps for other species to colonize, and therefore to increase diversity. EGR plants were partially planted and partially seeded (2014), and the species were chosen from nearby disturbed herbaceous communities and dry swards from peri-urban and rural areas [17]. After the plant establishment (two years), the green roof plant community was progressively dominated by *Sedum album*, thus other species were seeded in all the boxes in 2016, while some species self-colonized.

From November 2019 to December 2020, the vegetation and substrate temperature were monthly monitored in all EGR. The physical and chemical properties of the substrates were analyzed in December 2019. Plants were classified according to their life cycle, phenological, physiological characteristics, as an alternative to the species-based approach (plant functional types (PFTs), to highlight the ecological strategies. The species were grouped into the following PFTs: annual forbs, annual legumes, perennial forbs, perennial legumes, geophytes, CAM (i.e., *Sedum album* and *Sedum acre*), graminoids and bryophytes. *Sedum album* and *acre* can shift from C3 to CAM (Crassulacean Acid Metabolism) in case of water stress [18], so they are defined CAM in our PFTs (Table 1).

*2.2. Substrate Parameters*

The physical and chemical properties of the substrates were obtained as follows. Substrate samples were mashed and analyzed for pH ($H_2O$), conductivity (EC); organic carbon and $N_{tot}$ contents were determined by dry combustion using a Leco CHN Analyzer [19].

The substrate temperature was measured (from November 2019 to December 2020) in correspondence of each plant monitoring intervention, once a month, at noon, at three points along a diagonal in each box with a soil thermometer (total of 36 measurements). During the soil monitoring, the air temperature was also measured three times.

Bulk density (BD) and the tension curves of the substrates (VC, VPC and VP mixtures) were obtained using the UNI-EN method [20]. De Boodt and Verdonck [21] introduced the concept of water availability, expressed air capacity (AC), available water (AW), easily available water (EAW), and water buffering capacity (WBC). The nonlinear equation proposed by van Genuchten [22] to fit the retention data collected for several container media was applied by Milks et al. [23] and Wallach et al. [24] to container substrates. This equation correlates the water content, $\theta$ ($cm^3$ $cm^{-3}$), with the water suction 'h' (kPa; also expressed as 'pressure head'). A few parameters are needed (i.e., '$\alpha$', $\theta s$, $\theta r$, 'n' and 'm', where $\theta s$, $\theta r$ are the volumetric water contents at saturation and residual, respectively, '$\alpha$', 'n', 'm' are the parameters of the van Genuchten equation) which were determined by curve-fitting techniques using the RETC (RETention Curve) computer program code for quantifying the hydraulic functions of unsaturated soils [25].

**Table 1.** Species composition of the green roof. The plant functional types (PFTs) were assigned according to Pignatti et al. [26] and Pérez-Harguindeguy et al. [27] and the strategy type only for the species present in Klotz et al. [28]. AF = annual forbs; AL = annual legumes; PL = perennial legumes; BR = bryophytes; PF = perennial forbs; G = geophytes; GR = gramineae C = competitive; S = stress tolerator; R = ruderal (published in Vannuccchi et al. [10]).

| Species | Family | PFT | Growth Forms | Strategy Type | Introduction in the GR |
|---|---|---|---|---|---|
| *Allium roseum* L. | Amaryllidaceae | G | Bulbous | - | Planted 2014 |
| *Alyssum alyssoides* (L.) L. | Brassicaceae | AF | Scapose/rosulate | SR | Planted 2014 |
| *Anthyllis vulneraria* L. | Fabaceae | PL | Scapose/hemirosette | CSR | Planted 2014 |
| *Blackstonia 4olonizing* (L.) Huds. | Gentianaceae | AF | Scapose/rosulate | SR | Seeded 2016 |
| *Calendula arvensis* (Vaill.) L. | Asteraceae | AF | Scapose/hemirosette | R | Planted 2014/ seeded 2016 |
| *Centranthus macrosiphon* Boiss. | Caprifoliaceae | AF | Scapose | - | Seeded 2016 |
| *Crepis bursifolia* L. | Asteraceae | PF | Scapose | - | Spontaneously colonising |
| *Dianthus deltoides* L. | Caryophyllaceae | PF | Caespitose/rosulate | CSR | Planted 2014 |
| *Erodium cicutarium* (L.) L'Hér. | Geraniaceae | AF | Caespitose/scapose/hemirosette | R | Planted 2014 |
| *Geranium 4olo* L. | Geraniaceae | AF | Scapose/hemirosette | R | Planted 2014 |
| *Hypochaeris radicata* L. | Asteraceae | PF | Rosette | CSR | Seeded 2016 |
| *Lobularia maritima* (L.) Desv. | Brassicaceae | PF | Scapose/rosulate/hemirosette | SR | Seeded 2016 |
| *Muscari comosum* (L.) Mill. | Asparagaceae | G | Bulbous/rosulate | CSR | Planted 2014 |
| *Ornithogallum umbellatum* L. | Asparagaceae | G | Bulbous/rosulate | CSR | Planted 2014 |
| *Petrorhagia saxifraga* (L.) Link | Caryophyllaceae | PF | Caespitise/rosulate | CS | Planted 2014 |
| *Poa annua* L. | Poaceae | GR | Caespitose/hemirosette | R | Spontaneously colonising |
| *Portulaca oleracea* L. | Portulaccaceae | AF | Succulent scapose | | Spontaneously colonising |
| *Scrophularia peregrina* L. | Scrophularieae | AF | Scapose | - | Seeded 2016 |
| *Sedum acre* L. | Crassulaceae | CAM | Succulent | S | Planted 2014 |
| *Sedum album* L. | Crassulaceae | CAM | Succulent | S | Planted 2014 |
| *Senecio vulgare* L. | Asteraceae | AF | Scapose | | Spontaneously colonising |
| *Silene gallica* L. | Caryophyllaceae | AF | Scapose/rosulate | R | Planted 2014 |
| *Sochus oleraceous* L | Asteraceae | AF | | | Spontaneously colonising |
| *Trifolium arvense* L. | Fabaceae | AL | Scapose/rosulate/hemirosette | SR | Planted 2014 |
| *Trifolium campestre* L. | Fabaceae | AL | Scapose/rosulate/hemirosette | R | Planted 2014 |
| *Verbascum blattaria* L. | Scrophulariaceae | AF | Scapose/hemirosette | C | Seeded 2016 |
| *Mosses (Bryophyta)* | | BR | | | Spontaneously colonising |

### 2.3. Plant Community Composition and Structure

From November 2019 to December 2020, the number of individuals per species was counted every month in a 50 × 50 cm square at the central part of each box with a point frame for botanical surveying composed of ten pins (pin length: 46 cm distance between outer pins: 5 cm) (NHBS Ltd., Totnes, United Kingdon). The plant functional type (PFT) contributions (%) were calculated as the ratio between the number of PFTs touched by the pin and the total number of plants touched [26]. A total of 0.5 hits were assigned to PFTs, or species present but not touched [29]. PFT data were reported monthly as the total average, for the duration of the experiment.

Biodiversity indices were calculated: the Shannon diversity index ($H'$) [30], and the evenness of species ($J$) [31], as follows: $H' = \sum_{i=1}^{k} p_i log p_i$, where $k$ is the species number, and $p_i$ is the fraction of individuals belonging to the $i^{th}$ species; $J = H'/lnk$ where $H'$ is the Shannon diversity index, and $k$ is the species number. In addition, Simpson's index of dominance ($D$) [32] was calculated to assess the probability that two individuals randomly

selected from a sample belonged to the same species. $D = \sum \left( n/N \right)^2$ where $n$ is the number of individuals of a species, and $N$ the total number of individuals of all species.

*2.4. Statistical Analysis*

Statistical analysis was conducted using open-source R software (version 4.1.1). PFT contributions and biodiversity indices (Shannon, evenness and Simpson indices) were monitored over time and compared in different substrates and seasons. The data collected were subjected to the parametric ANOVA test (for homoscedastic normally distributed populations), non-parametric ANOVA, Kruskall–Wallis non-parametric ANOVA test (for homoscedastic non-normally distributed populations) and Friedman's rank sum test (for heteroscedastic non-normally distributed populations). The homogeneity of variance within populations was verified with Bartlett's test and the Gaussian distribution with the Shapiro–Wilk normality test. Post hoc comparisons between groups were carried out with the paired sample *t*-test (or Mann–Whitney non-parametric U-test for non-normally distributed populations), using the Bonferroni adjustment for multiple comparison correction. In all the tests, a *p*-value of $p < 0.05$ was used as the threshold of statistical significance. A principal component analysis was performed to visualize the distribution of substrates in the plane identified by the first two main components of the space generated by the PFTs (annuals, CAM, geophytes, and bryophytes), substrate temperature (temp), Shannon index (H) and available water (AW).

## 3. Results

### 3.1. Substrate Parameters

Substrate properties are reported in Table 2. The only parameter significantly different among substrates was $N_{tot}$, which was lower in VP compared to the substrates containing compost: VC and VPC. The temperature in the substrates during the experiment was lower than the air temperature in all seasons, which in summer tended to be 8–10 °C lower (Figure 1). Generally, significant differences amongst substrates were observed in relation to average substrate temperature. In fact, VP showed a higher temperature than VC ($p = 0.026$), while no differences were observed between VP and VPC or between VC and VPC.

**Table 2.** Chemical and physical properties of the substrates (* = data published by Vannucchi et al. [15]).

|     | pH * | EC * | C$_{org}$ * | N$_{tot}$ * | BD |
| --- | --- | --- | --- | --- | --- |
|     | H$_2$O | dS/m | % | % | g cm$^{-3}$ |
| VC | 8.0 ± 0.06 | 0.2 ± 0.03 | 5.8 ± 1.08 | 0.65 ± 0.15 a | 0.5 ± 0.01 |
| VPC | 8.2 ± 0.14 | 0.2 ± 0.02 | 3.5 ± 0.37 | 0.36 ± 0.05 a | 0.66 ± 0.01 |
| VP | 8.5 ± 0.05 | 0.2 ± 0.01 | 3.8 ± 0.42 | 0.26 ± 0.01 b | 0.68 ± 0.15 |

EC = electrical conductivity; C$_{org}$ = organic carbon; N$_{tot}$ = total nitrogen, BD = Bulk Density. VC = vulcaflor + compost; VPC = vulcaflor + paper sludge pellet + compost; VP = vulcaflor + paper sludge pellet. Data are means of 4 replicates ± SD. Different letters show a statistical difference for $p < 0.05$ between substrates. * published in Vannuccchi et al. [10].

VC showed an increase in water content compared to VPC and VP. The highest 'θs' was shown by VC, while VP and VPC showed a lower water content, due to their pellet content (Table 3). Approximately 25% of the water of saturated VC was lost when suction was increased from 0 to 1 kPa. As the suction was increased further, the water loss was less drastic, and at the highest suction applied (h = 10 kPa), the medium still held more than 24% water by volume. For VP, the water loss in the 0 to 1 kPa range was very sharp (about 50%), and the volume of water held at h = 10 kPa was only slightly lower than 20% of volume. VPC behaved between VC and VP except in the 0 to 1 kPa range (Figure 2). As the water loss in the 0 to 1 kPa range and AC are inversely correlated, AC increased correspondingly. Figure 2 presents the nonlinear least-squares fit of the van Genuchten equation along with the data observed. Of the five parameters in the model, only 'θs' was

measured, whereas '$\alpha$', '*n*', and '$\theta$r' were calculated; 'm' was set equal to $m = 1-1/n$ to fit the measured data (Table 3), as suggested by Wallach et al. [24] and da Silva et al. [33].

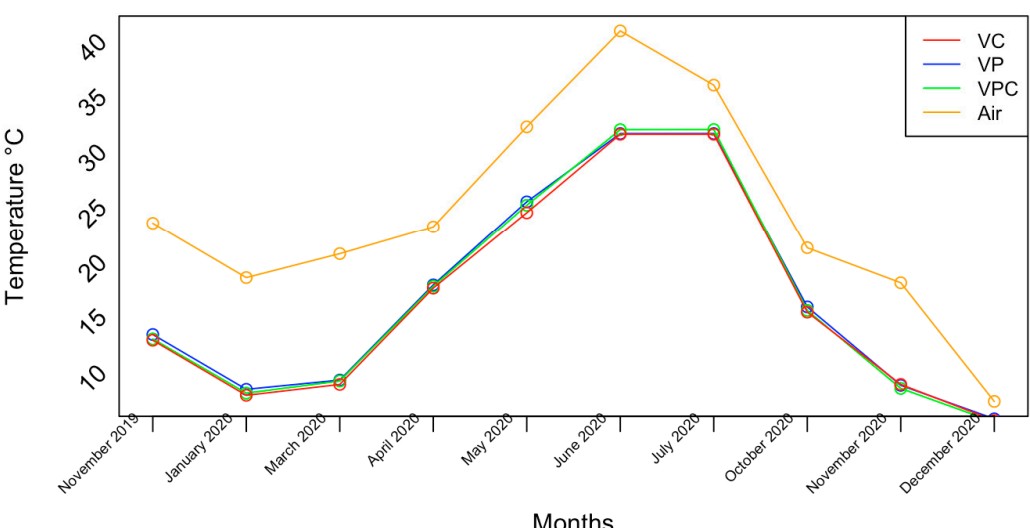

**Figure 1.** Temperature of the air and the substrates from November 2019 to December 2020 (months not monitored are not reported). Circles show the months with significant differences between substrate temperature distributions ($p < 0.05$). VC = vulcaflor + compost mixture; VPC = vulcaflor + paper sludge pellet + compost mixture; VP = vulcaflor + paper sludge pellet mixture.

**Table 3.** Van Genuchten equation parameters '$\alpha$', '*n*', and '$\theta$r' calculated via RETC, fixing $m = 1-1/n$. '$\theta$s' ($cm^3$ $cm^{-3}$) are measured data. $R^2$ reports the coefficients of determination of the nonlinear regression. Nomenclature: VC = vulcaflor + compost mixture; VPC = vulcaflor + paper sludge pellet + compost mixture; VP = vulcaflor + paper sludge pellet mixture. Air capacity (AC), available water (AW), easily available water (EAW), and water buffering capacity (WBC).

| Parameter | Unit | VC | VPC | VP |
|---|---|---|---|---|
| $\theta$s | $cm^3$ $cm^{-3}$ | $0.79 \pm 0.003$ | $0.75 \pm 0.009$ | $0.73 \pm 0.005$ |
| $\theta$r | $cm^3$ $cm^{-3}$ | $0.232 \pm 0.027$ | $0.192 \pm 0.009$ | $0.189 \pm 0.003$ |
| $\alpha$ | $kPa^{-1}$ | $0.094 \pm 0.005$ | $0.512 \pm 0.081$ | $0.206 \pm 0.012$ |
| *n* | - | $2.7996 \pm 0.642$ | $1.8965 \pm 0.110$ | $2.488 \pm 0.131$ |
| $R^2$ | - | 0.999 | 0.999 | 0.999 |
| AC | % *v/v* | $16.8 \pm 2.1$ | $42.1 \pm 1.7$ | $36.4 \pm 2.1$ |
| AW | % *v/v* | $38.0 \pm 1.6$ | $11.6 \pm 2.0$ | $16.1 \pm 0.7$ |
| EAW | % *v/v* | $36.0 \pm 1.4$ | $9.1 \pm 2.3$ | $15.3 \pm 0.8$ |
| WBC | % *v/v* | $2.0 \pm 0.2$ | $2.5 \pm 0.3$ | $0.8 \pm 0.1$ |

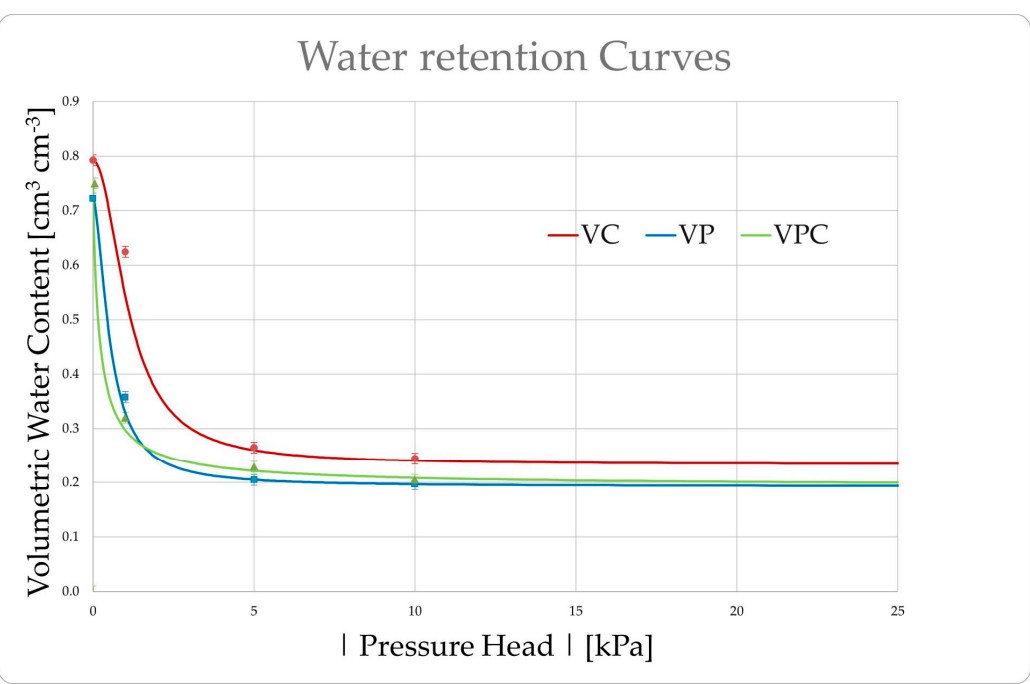

**Figure 2.** Measured (symbols) and fitted (lines) water retention curves of the tested media. Symbols and nomenclature: VC = Vulcaflor + compost mixture; VPC = Vulcaflor + paper sludge pellet + compost mixture; VP = Vulcaflor + paper sludge pellet mixture. Standard deviations of experimental water retention points are reported as bars.

*3.2. Plant Community Composition and Structure*

The composition and the structure of the plant community in each substrate are reported in Figures 3 and 4. In terms of PFTs, geophytes, annual forbs and legumes did not differ amongst substrates. Geophytes were 8.3, 6.1 and 5.8% in VC, VPC and VP, respectively. Annual forbs were 12.8% in VC, 13.2 in VPC and 20.6 in VP. Annual legumes were 1.1, 2.3 and 1.7% in VC, VPC and VP, respectively. CAM was the most common functional type in the plots and significantly higher in VC (76.1%, *p*-value of VP vs. VC with Bonferroni correction is 0.003) and VPC (74.1%, *p*-value of VP vs. VPC with Bonferroni correction is 0.003), followed by VP (60.8%). Bryophytes were significantly higher in VPC (4.1%) than in VP (1%, *p*-value of VP vs. VPC with Bonferroni correction is 0.003) and VC (1.8%, *p*-value of VC vs. VPC with Bonferroni correction is 0.005) (Figure 3). The Shannon index was significantly higher in VP (1.30, *p*-value of VP vs. VC with Bonferroni correction is 0.002) and in VPC (1.21, *p*-value of VPC vs. VC with Bonferroni correction is 0.03) followed by VC (1.06), whereas the Simpson index reached significantly lower values in VP (0.40), followed by VPC (0.45) and VC (0.54), and all values resulted as being significantly different with Bonferroni correction. The evenness was significantly higher in VP (0.61, *p*-value of VP vs. VC with Bonferroni correction is 0.015) and VPC (0.59, *p*-value of VPC vs. VC with Bonferroni correction is 0.032) than in VC (0.50). The biplot of the PCA analysis, obtained using the first two PCs (Figure 5), highlighted that the Shannon diversity index (H) positively correlated with the annual plant type. A negative correlation between H, annuals and CAM occurred. In addition, H and annuals were related to VP and VPC substrates, while VC correlated with CAM and AW.

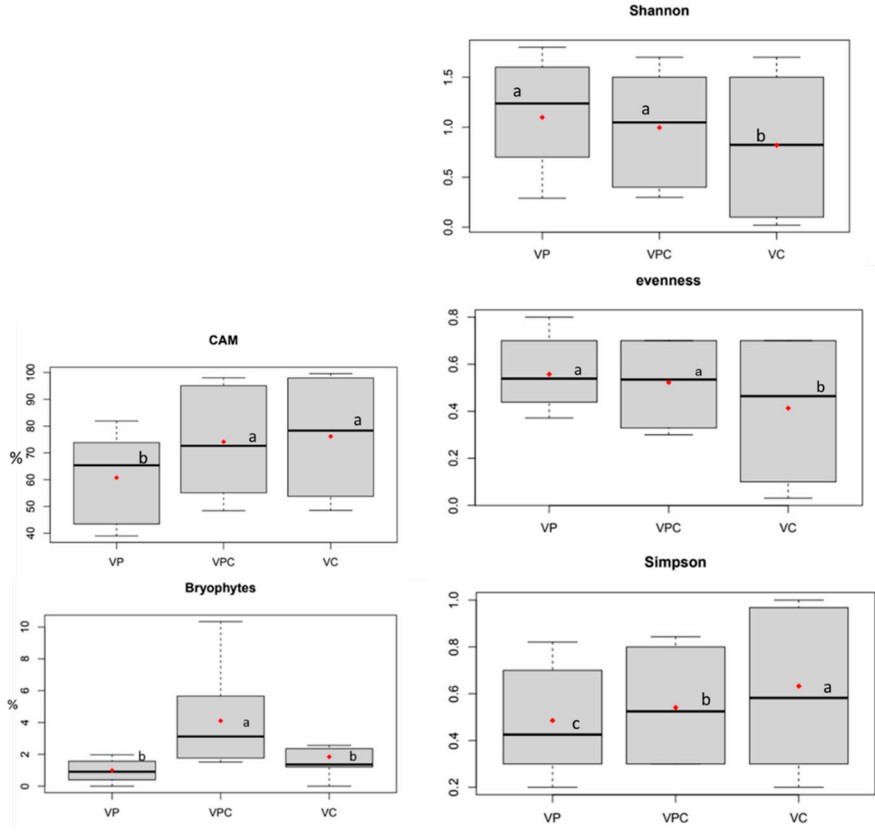

**Figure 3.** Boxplots of the plant community composition and structure, in terms of functional type cover (%) and biodiversity indices, monitored in the substrates from November 2019 to December 2020, a diamond represents the average, a line represents the median. VC = Vulcaflor + compost mixture; VPC = Vulcaflor + paper sludge pellet + compost mixture; VP = Vulcaflor + paper sludge pellet mixture; CAM = crassulacean acid metabolism (sedums). Data are means of 4 replicates ± SD. Different letters show a statistical difference for $p < 0.05$ between substrates. The red dots represent the mean values.

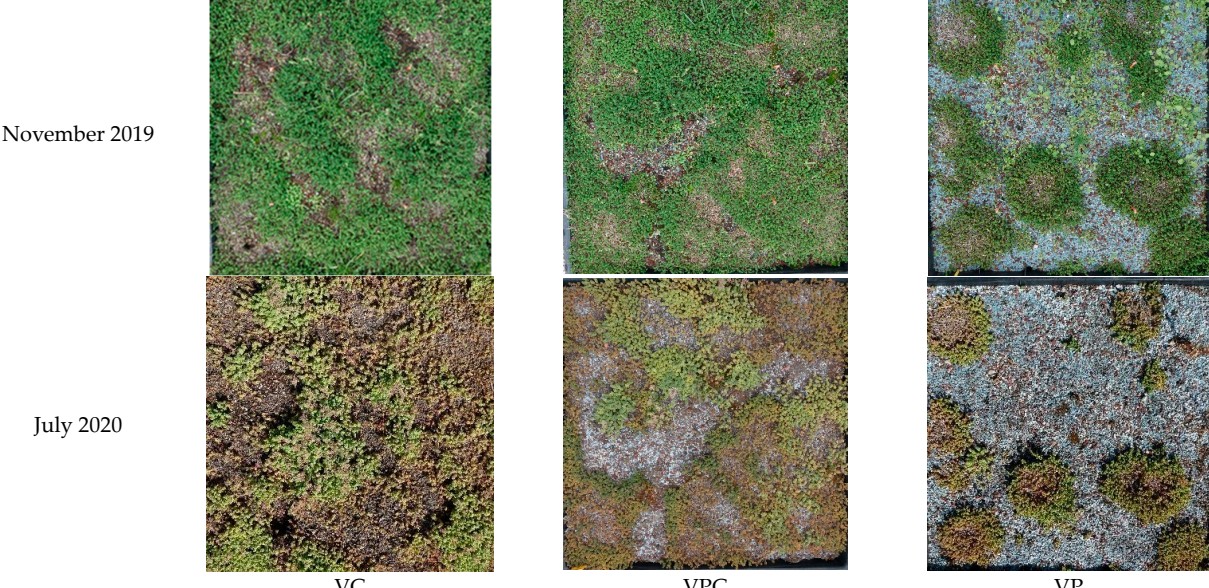

**Figure 4.** Examples of the EGR boxes in autumn (November 2019) when rain allows the seedling recruitment of annuals, especially in VP where empty spaces are larger than in VC and VPC. In summer

(July 2020), due to drought, only the sedum survives. VC = Vulcaflor + compost; VPC = Vulcaflor + paper sludge pellet + compost; VP = Vulcaflor + paper sludge pellet.

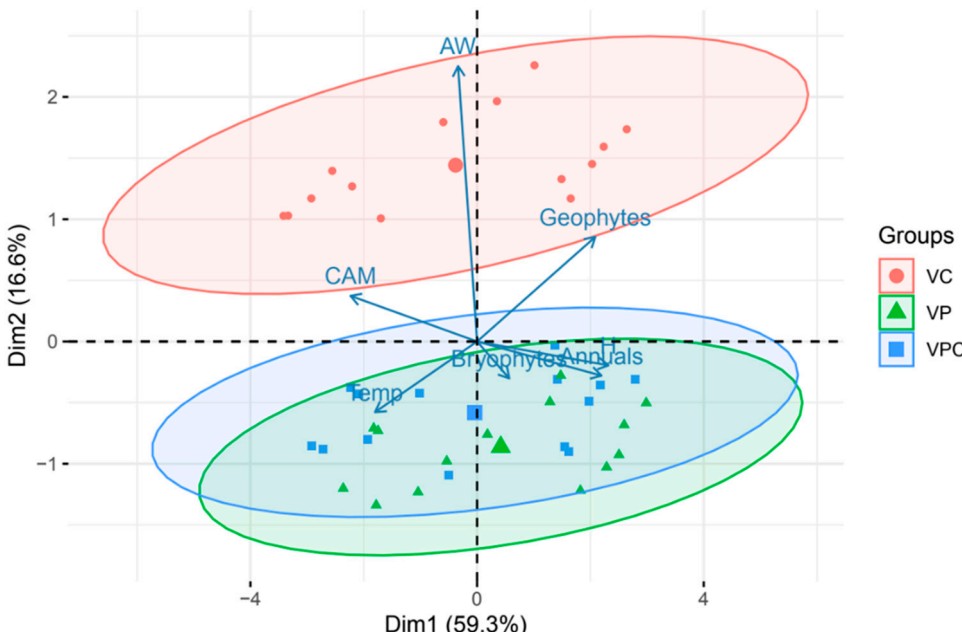

**Figure 5.** PCA of substrate temperature (temp), PFTs (Annuals, including forbs and legumes, CAM, geophytes, and bryophytes), Shannon index (H) and available water (AW) detected in the substrates. VC = Vulcaflor + compost mixture; VPC = Vulcaflor + paper sludge pellet + compost mixture; VP = Vulcaflor + paper sludge pellet mixture.

## 4. Discussion

The importance of green roofs in providing ecosystem services has been widely investigated, especially in terms of mitigating the temperatures, roof surface insulation and water retention [4], and potentially to increase biodiversity through the creation of habitats for urban flora and fauna [34,35]. The best trade-off between substrate temperature mitigation and the improvement of plant diversity still deserves investigation.

In our study, the use of recycled organic materials, i.e., paper sludge (PS) and compost, in different percentages in EGR substrates, evidenced different behaviors in terms of temperature, water retention and biodiversity.

The green roof performance strongly depends on its substrate hydraulic and thermal properties and moisture content because they affect the heat and fluid flow [36–38]. In our study, significant differences in average substrate temperature were observed amongst substrates. The VPC substrate provided annual average temperature similar as VC, while the substrate with a higher percentage of pellet (VP) showed a higher temperature value compared to the others. All the EGR substrates, although with no irrigation and just the sparse monolayer of sedum cover, showed a notable reduction (8–10 °C) compared to the air temperature, which in summer exceeded 40 °C. The differences between substrate temperature may be attributable to the granulometry, as pellet and tephra are highly porous, enabling the warming of the media [6]; on the other hand, VC had a more widespread vegetation in all seasons, which contributes to reduce the irradiation on the surface [8,12]. In cities with very high summer temperatures, the substrates of cultivated roofs are on average 2–3 °C lower than uncultivated roofs [39]. Unlike in other studies, the dominance of sedum (CAM) in the EGR substrate did not attenuate the substrate temperature of the most vegetated (VC), compared to the less vegetated (VP) in the summer [40]. Sedums, which were the most common species in compost substrate, could act as a nurse crop to neighboring plants by reducing the summer substrate temperature [41]. Conversely, in

this experiment, the summer substrate temperatures were very similar between the trials despite the different vegetation.

The hydraulic properties of the tested substrates indicated that the presence of PS in the substrate enhanced the air content, while compost conferred high-water capacity to the substrate, enabling the green roofs to mitigate the runoff volume. In fact, the substrate with a high amount of compost (VC) showed a higher capacity to store water (AW, EAW, WBC) than VPC and VP. The PS substrates (VPC and VP) showed a higher air capacity and limited easily available water compared to the VC substrate [42]. This was also confirmed by the combination of Van Genuchten Equation with a predictive model for the unsaturated hydraulic conductivity in a combined hydraulic model [33]. The high-determination coefficients of the nonlinear regression ($R^2 > 0.99$) indicated that the estimated drying curves were accurate for the tested EGR substrates. Despite the large capacity of water storage, compost can be highly hydrophobic once completely dry [43]; it increases the weight when full of water, and releases excess elements, especially N; it is also subject to shrinking and includes weed seeds. For these reasons, the percentage of compost amendment in EGR substrates should be limited to 15–35% maximum [44]. Mixing compost and PS in EGR can reduce the limitation derived by compost use.

The compost and PS revealed useful properties for the management of EGRs, also related to the differences in nitrogen content. Besides the lower water retention and higher air filling capacity, the use of PS greatly reduced the N content. The third component of the substrate, tephra, is a great amendment for green roofs confers low density, as it is light and porous and has wide exchange surfaces. On the other hand, it has low easily available water and is a non-renewable material so its cost may impact the realization [44]. The differences in nitrogen content affected the plant composition and diversity of the plant communities, in terms of plant functional types. The substrate composed of compost and tephra (VC), rich in nitrogen content, was dominated by CAM, and presented lower plant diversity. In the substrates containing PS (VP and VPC), the percentage of CAM was limited by the low nitrogen content. Decreasing the percentage of compost and increasing the PS the colonization by sedums (CAM) had been reduced, this improved the establishment of winter annuals and led to a biodiverse plant community in VPC and VP [10]. In particular, the VP substrate had a very low N content, similar to stressful edaphic conditions, which in natural vegetation would strongly limit the development of the biomass and promote the growth of stress tolerant species [45]. The species growing on the roof were not in competition with each other, due to the scarcity of resources in terms of nutrients and the shallow root space, thus the EGR communities remained in relatively early successional stages [34]. However, in line with the seasonal adaptation, CAM dominated the summer period, while geophytes, bryophytes and annuals developed from autumn to spring [46]. The different growth forms and complementary functional types in EGRs, increase the overall diversity [47]. The annual plant community self-regenerated in the rain, with no need for irrigation, and the seedling recruitment started from October–November. The presence of annuals drives the plant diversity in EGR, where there is space free for plant colonization [34]. On the other hand, sedum spread on the substrates with the highest amount of compost (VC) and thanks to its phenotypical plasticity can adapt to low fertility substrates, just reducing the spread of its canopy (Figure 4). At first drought, generally from May–June, depending on the yearly climate, the annuals dried out after producing seeds. In some cases, annual species seeded in green roofs can be irrigated at least at the first stages to allow seeds to germinate [48]. In this experiment the seeds germinated thanks to the wet and cool season rains, with no further irrigation, and simply following the Mediterranean natural cycles. EGR stressful conditions may worsen due to, amongst the factors, substrate shallow depth [47] and limited nutrient availability [3]. The highest plant diversity (Shannon and evenness indices) was related to the more stressed substrate (VP, VPC); in this study, the lack of nitrogen created the stressful conditions, while the compost substrate (VC) had the lowest plant diversity.

In climates like the Mediterranean, recommendations for the EGR application in purpose of the self-sustainability would be to encourage the naturalization of annual species among crassulaceans by adding, in the wet season, a mix of seeds, individuating the species from local spontaneous phytocoenosis. If the species are preferably entomogamous, the combination may result in a novel mix, which would be effective in offering a habitat to pollinators [49]. Moreover, the adding of PS (15–20%) as an amendment can be carried out, mixed with compost and tephra or other light material for a depth of 10 cm.

**5. Conclusions**

Our study shows that EGRs can become self-sustainable in a Mediterranean climate by exploiting the seasonal complementarity of plant functional types (PFTs) (i.e., with CAM in the summer, and annual species in the cool and wet season), growing on substrates composed of recycling materials (compost and paper sludge). The development of PFTs and the biodiversity had been affected by the substrate composition as well as the substrate temperature, water/air capacity. The growing media composition influenced differently the EGR. The presence of a high percentage of compost (VC) in the substrate increased the water content and nitrogen but reduced the species diversity. The substrate composed of tephra and pellet (VP) showed high plant diversity but also high substrate temperature. Mixing tephra, pellet and compost (VPC) was the best trade-off in terms of the spread of vegetation and empty gaps, enabling temperature mitigation. However, in summer drought, the plant community cover and diversity were limited to CAM functional types in all substrates. The compost and paper sludge, employed as amendments in this research, showed good capacity to retain water and air and were almost equal in relation to temperature mitigation. In addition, the use of compost and paper sludge implies positive effects on the circular economy. Future research should be focused on the monitoring of annual plant community over time to explore the persistence to the species and the arrival of self-colonizers. An annual life form is a winning plant strategy to cope with the very dry summers of Mediterranean, which should be exploited to implement nature-based solutions in a sustainable urban green infrastructure and plant diversity.

**Author Contributions:** Conceptualization, F.B., F.V. and C.B.; methodology, F.B., C.C., F.V. and C.B.; software, C.C. and C.B.; validation, F.B., F.V. and C.B; investigation, F.B.; data curation, F.B., F.V. and C.B.; writing—original draft preparation, F.B. and F.V.; writing—review and editing, F.B., F.V. and C.B.; visualization, F.B., C.C. and F.V.; funding acquisition, F.B. All authors have read and agreed to the published version of the manuscript.

**Funding:** This research was funded by Fondazione Cassa di Risparmio di Lucca, grant number CARPELLET (ID 24279) and by the National Recovery and Resilience Plan (NRRP), Mission 4 Component 2 Investment 1.4—Call for tender No. 3138 of 16 December 2021, rectified by Decree No. 3175 of 18 December 2021 of Italian Ministry of University and Research funded by the European Union—Next Generation EU; Project code CN_00000033, Concession Decree No. 1034 of 17 June 2022 adopted by the Italian Ministry of University and Research, CUPB83C22002930006, Project title "National Biodiversity Future Center—NBFC".

**Data Availability Statement:** Not applicable.

**Conflicts of Interest:** The authors declare no conflict of interest.

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
