# Peer review of "Mediterranean Extensive Green Roof Self-Sustainability Mediated by Substrate Composition and Plant Strategy"

_horticulturae, doi:10.3390/horticulturae9101117_

Round 1

Reviewer 1 Report (Previous Reviewer 2)

The manuscript titled "Composition and Temperature of Substrate and Plant Functional Types as Determinants of Mediterranean Extensive Green Roof Self-Sustainability," authored by Vannucchi et al., primarily addresses the following key objectives:

1. To evaluate the impact of incorporating recycled materials (compost and paper sludge) into extensive green roof (EGR) substrates on the promotion of self-sustaining vegetation.

2. To identify the most suitable EGR substrate composition for Mediterranean climates, considering the balance between temperature regulation and plant diversity. This investigation encompasses an analysis of water retention, substrate temperature, and their effects on vegetation dynamics, including plant diversity, abundance, and functional types.

This study is thoughtfully designed and provides substantial evidence to support its conclusions. While it may not introduce groundbreaking innovations to the field, the data it presents contributes to expanding the possibilities for green roof development. Here are some minor comments to enhance the manuscript's quality:

The title appears somewhat lengthy for the straightforward nature of the study. Consider a more concise title.

In the Conclusions section, when discussing how substrate composition affects substrate temperature, it would be beneficial to offer potential explanations for this phenomenon. Additionally, it would be beneficial to cite relevant prior studies for these explanations to provide context.

In Figure 1, there are dotted lines present in the figure legend, but none are visible in the plot itself. This discrepancy may have occurred during format conversion. It is advisable to thoroughly review all figures and tables after any format conversion to ensure accuracy.

Author Response

Reviewer 1

The manuscript titled "Composition and Temperature of Substrate and Plant Functional Types as Determinants of Mediterranean Extensive Green Roof Self-Sustainability," authored by Vannucchi et al., primarily addresses the following key objectives:

  1. To evaluate the impact of incorporating recycled materials (compost and paper sludge) into extensive green roof (EGR) substrates on the promotion of self-sustaining vegetation.
  2. To identify the most suitable EGR substrate composition for Mediterranean climates, considering the balance between temperature regulation and plant diversity. This investigation encompasses an analysis of water retention, substrate temperature, and their effects on vegetation dynamics, including plant diversity, abundance, and functional types.

This study is thoughtfully designed and provides substantial evidence to support its conclusions. While it may not introduce groundbreaking innovations to the field, the data it presents contributes to expanding the possibilities for green roof development. Here are some minor comments to enhance the manuscript's quality:

The title appears somewhat lengthy for the straightforward nature of the study. Consider a more concise title. We have modified the title.

In the Conclusions section, when discussing how substrate composition affects substrate temperature, it would be beneficial to offer potential explanations for this phenomenon. Additionally, it would be beneficial to cite relevant prior studies for these explanations to provide context. We added an explanation with references in the discussion, and in Conclusions.

In Figure 1, there are dotted lines present in the figure legend, but none are visible in the plot itself. This discrepancy may have occurred during format conversion. It is advisable to thoroughly review all figures and tables after any format conversion to ensure accuracy. We have modified the figure 1 accordingly.

Reviewer 2 Report (Previous Reviewer 3)

The review of the manuscript titled "Composition and temperature of the substrate and plant functional types are keys to the self-sustainability of Mediterranean extensive green roofs"

This is a review of the manuscript titled "Composition and temperature of the substrate and plant functional types are keys to the self-sustainability of Mediterranean extensive green roofs "after its resubmission to the Horticulturae (MDPI).

First, why are the row numbers not provided? It is extremely inconvenient when making a review. Second, not all the reviewer’s comments have been properly answered. For example, L223. I suggest making the Y-axis equal to 100% in all panels.

Specific comments:

Abstract. It is unclear in the sentence "The VPC and VC had the same average substrate temperature, with values lower than VC." the values are the same or lower. The last sentence of that paragraph is too long and has an unclear structure. Try to divide it into two sentences.

Introduction. The third paragraph has poor structure; the discussion of substrate type mixes with the physical parameters. I suggest improving that.

2.1 Experimental design. In the sentence "The initial planting (2014) was partially planted and partially seeded, and the species were chosen from nearby disturbed herbaceous communities and dry swards from peri-urban and rural areas" planting was planted? What do you mean by this?

In the sentence "From November 2019 to December 2020, the vegetation and substrate properties (temperature and water retention) in EGR were monitored.", with what frequency did you monitor the parameters? At all plots?

Fig. 2. Could you provide the standard errors for the parameters?

Author Response

Reviewer 2The review of the manuscript titled "Composition and temperature of the substrate and plant functional types are keys to the self-sustainability of Mediterranean extensive green roofs"

This is a review of the manuscript titled "Composition and temperature of the substrate and plant functional types are keys to the self-sustainability of Mediterranean extensive green roofs "after its resubmission to the Horticulturae (MDPI).

First, why are the row numbers not provided? It is extremely inconvenient when making a review. We apologize for the inconvenience: we added row numbers.

Second, not all the reviewer’s comments have been properly answered. For example, L223. I suggest making the Y-axis equal to 100% in all panels. We decided to show different Y axes values to better visualize the differences amongst substrates.

Specific comments:

Abstract. It is unclear in the sentence "The VPC and VC had the same average substrate temperature, with values lower than VC." the values are the same or lower. The last sentence of that paragraph is too long and has an unclear structure. Try to divide it into two sentences. The abstract has been modified accordingly.

Introduction. The third paragraph has poor structure; the discussion of substrate type mixes with the physical parameters. I suggest improving that. We have improved the paragraph in lines 56-84.

2.1 Experimental design. In the sentence "The initial planting (2014) was partially planted and partially seeded, and the species were chosen from nearby disturbed herbaceous communities and dry swards from peri-urban and rural areas" planting was planted? What do you mean by this? We modified the sentence.

In the sentence "From November 2019 to December 2020, the vegetation and substrate properties (temperature and water retention) in EGR were monitored.", with what frequency did you monitor the parameters? At all plots? We clarified the sentence accordingly.

Fig. 2. Could you provide the standard errors for the parameters? The standard errors of the parameters used for the curves are reported in table 3. We added this information in the figure 2 caption.

Round 2

Reviewer 2 Report (Previous Reviewer 3)

The review of the manuscript titled "Mediterranean extensive green roof self-sustainability mediated by substrate composition and plant strategy"

This is a second review of the manuscript titled "Mediterranean extensive green roof self-sustainability mediated by substrate composition and plant strategy" after its resubmission to the Horticulturae (MDPI).

The manuscript was elaborated but still needs some work. For example, the Discussion section is still a bit inconsistent and lacks logic in the narration. Now it is a mixture of different pieces without a logical connection. And provide a finalizing sentence for the section, for example, recommendations for the EGR application. The Conclusions are poorly structured. The sentence at L391-393 can be the first sentence of the Conclusions section. L385-391 should be introduced in the middle of the section as a detailed findings of the research.

Specific comments:

L80-83. This sentence repeats the sentence at L96-98.

L118-122. Explain the differences in the Vulcaflor amount among variants. Why did you use these different ratios?

L243. Explain the air capacity (AC).

L251. Standard errors for the precise values in Fig. 2 are not reported in Table 3!

L276. Fig. 3. Why are only Bryophytes and CAM presented among all PFTs?

L289. Fig. 5. Why did you reduce forbs and legumes to “Annuals” in PCA? Explain. What was the reason?

L295-303. Why was only the CAM group discussed here? What about other PFTs?

L330-347. This paragraph sounds like it should be moved to the first paragraph of the Discussion section.

Author Response

The review of the manuscript titled "Mediterranean extensive green roof self-sustainability mediated by substrate composition and plant strategy"

This is a second review of the manuscript titled "Mediterranean extensive green roof self-sustainability mediated by substrate composition and plant strategy" after its resubmission to the Horticulturae (MDPI).

The manuscript was elaborated but still needs some work. For example, the Discussion section is still a bit inconsistent and lacks logic in the narration. Now it is a mixture of different pieces without a logical connection. And provide a finalizing sentence for the section, for example, recommendations for the EGR application. We have modified the discussion and added EGR recommendations at the end of the section according to suggestions. Thank you for your valuable comments.

The Conclusions are poorly structured. The sentence at L391-393 can be the first sentence of the Conclusions section. L385-391 should be introduced in the middle of the section as a detailed findings of the research. We have modified the conclusions according to suggestions. Thanks for your valuable comments.

Specific comments:

L80-83. This sentence repeats the sentence at L96-98. The sentence in L96-98 has been deleted.

L118-122. Explain the differences in the Vulcaflor amount among variants. Why did you use these different ratios? We specified the criteria used for the composition of the different substrates in the Experimental design section (L116-118).

L243. Explain the air capacity (AC). We explained the AC in L232-234.

L251. Standard errors for the precise values in Fig. 2 are not reported in Table 3! We added the standard deviation as bars in the figure 2. We apologize for the misunderstanding, and we thank the reviewer for the opportunity for clarification.

L276. Fig. 3. Why are only Bryophytes and CAM presented among all PFTs? We reported in figure 3 only the PFTs that showed significant differences amongst substrates. The other PFTs are reported in the text (L252-261).

L289. Fig. 5. Why did you reduce forbs and legumes to “Annuals” in PCA? Explain. What was the reason? We thank the reviewer for the opportunity for clarification. As annual legumes were below 2% and annual forbs was the most representative (reaching 20%), we grouped annual legumes and annual forbs to better assess the effect of substrates on the PFTs.

L295-303. Why was only the CAM group discussed here? What about other PFTs? We focused on CAM as this PFT was the most common functional type in the plots, as mentioned in the results (L256-257).

L330-347. This paragraph sounds like it should be moved to the first paragraph of the Discussion section. We have modified the conclusions.

This manuscript is a resubmission of an earlier submission. The following is a list of the peer review reports and author responses from that submission.

Round 1

Reviewer 1 Report

The ms of Vannucchi et al. describes an experiment with different substrate composition for green roofs to evaluate their influence on water retention, substrate temperature and dynamics of vegetation composition. Despite authors present methodically competent work and a suitable statistical analysis I have several substantial questions to this study. For instance, authors should present list of plant species. What species were planted, what species were seeded, what species were from nearby communities and what from rural areas? What species were self-colonized? Were there plant species with C4-type of photosynthesis? Were all the forbs and legumes C3 plants? What values are presented on the vertical axis in fig. 3 and 4 – are they abundance or cover? For species number or for plant functional type as a whole? However, referring to the link in Methods - Vannucchi et al.2022 (by the way, there are not Vannucchi et al. (2018, 2021) in References - there is only Vannucchi et al.2022) I found more important flaws. This ms is too similar in Methods, Results and Conclusions to paper of Vannucchi et al.2022 (Vannucchi, F., Buoncristiano, A., Scatena, M., Caudai, C., and Bretzel, F. (2022). Low productivity substrate leads to functional diversification of green roof plant assemblage. Ecological Engineering, 176, 106547). The carried-out experiment is the same. Moreover, in this ms the most part of results is identical to Vannucchi et al.2022, namely table 1, Fig.3, Fig.4, Fig.6 present the same results. Fig. 7 (PCA analysis) presents the same analysis on the same data only in some different way (with different grouping). The main conclusions of this ms do not differ from previous paper of Vannucchi et al.2022. I think this ms should be rejected.

Reviewer 2 Report

To increase the impact of this study, the author should provide a rational discussion about what can be more sources of V, P, and C, especially waste-based sources. These details will allow researchers worldwide to optimize the ratio of V/P/C based on exciting environmental parameters.

For Fig 3, 4, and 6 - please provide the statistical analysis (p-values) that can help the readers to check which differences are significant.

Reviewer 3 Report

The review of the manuscript titled "Substrate composition, temperature and plant functional types are key to the self-sustainability of extensive green roofs"

The manuscript studies how the compost and deinked paper sludge influence water retention, substrate temperature attenuation, and plant diversity in an EGR experiment in the Mediterranean climate. The first issue that arises concerns the title. Which temperature do you mean? Air? Substrate? All the factors are key factors with the same contribution? I recommend reconsidering the title and introducing the location.

EGR is described in the introduction as an environmentally friendly solution for the Mediterranean climate to mitigate the challenges posed by climate change. The authors define the main components and parameters of EGR. However, it is not easy to understand the novelty/originality of the research. I would suggest highlighting the novelty of the study if it exists. In this section, I recommend reformulating the aims by placing the EGR in the first place of the sentence. Moreover, I suggest adding the research hypotheses concerning the better substrates for EGR in a Mediterranean climate.

The methods section is well written, including the experimental design and data analysis. Some questions are about the PCA parameters chosen, and they need some explanation. If you choose more parameters, will the PCA graph be the same? In Section 2.4, I suggest adding the definitions of PFTs to make it clear for the reader.

The Results section does not provide the findings on the key parameters for the EGR. For Fig. 7, I suggest building two PCAs: one for PFTs and diversity indices and another for environmental parameters. Perhaps it will help to define the key parameters for EGR.

In the discussion part, paragraph in the L284–296 is better moved to the introduction since it discusses different substrates and their properties, some of which do not relate to the study. This section should be rewritten in the following way: first, you discuss the obtained results, and then you compare them to the published ones. Besides, the narration should be more logical and go smoothly.

Conclusions should be rewritten. At the moment, they do not provide clear recommendations (pros and cons) in terms of plant diversity and environmental conditions for EGR usage in the Mediterranean climate.

Some references are not provided. Examine all of the text. 26% of the references are older than 10 years.

Here, are some specific comments.

L20. Correct "of" in the part of the sentence that says "perennials, legumes, of, geophytes) were".

L40. Check the reference. It is not provided.

L60. Here, you can use the previously introduced abbreviation "EGR".

L73. Add some relevant references.

L86. Vannucchi et al. (2021) is not provided.

L110. Correct to "substrate temperature".

L114. Why are the substrates growing?

L120. Correct the units in the brackets (cm3.cm-3).

L160-163. Why aren't all PFTs represented? Earlier, you described more PFTs, for example, at L104–106. The same question applies to the parameters studied (air temperature, Ntot, bulk density, different types of water availability, etc.).

L170. Refer to Fig. 1.

L173-177. Check the figure 2 reference here.

L223. I suggest making the Y-axis equal to 100% in all panels.

L290. Remove the unnecessary comma.

L372. What do you mean by "the water/air content"?
